# Sphingosine Phosphate Lyase Is Upregulated in Duchenne Muscular Dystrophy, and Its Inhibition Early in Life Attenuates Inflammation and Dystrophy in Mdx Mice

**DOI:** 10.3390/ijms23147579

**Published:** 2022-07-08

**Authors:** Anabel S. De la Garza-Rodea, Steven A. Moore, Jesus Zamora-Pineda, Eric P. Hoffman, Karishma Mistry, Ashok Kumar, Jonathan B. Strober, Piming Zhao, Jung H. Suh, Julie D. Saba

**Affiliations:** 1Department of Pediatrics, University of California San Francisco, 550 16th Street, Box 0110, San Francisco, CA 94143, USA; asdelagarza@gmail.com (A.S.D.l.G.-R.); jzamorapineda@luc.edu (J.Z.-P.); mistryk5@gene.com (K.M.); ashok.biochemistry@aiimsbhopal.edu.in (A.K.); piming.zhao@renoviron.com (P.Z.); junghyuk@mac.com (J.H.S.); 2Senator Paul D. Wellstone Muscular Dystrophy Specialized Research Center, Department of Pathology, The University of Iowa Carver College of Medicine, Iowa City, IA 52242, USA; steven-moore@uiowa.edu; 3Department of Pharmaceutical Sciences, School of Pharmacy and Pharmaceutical Sciences, Binghamton University-State University of New York, Binghamton, NY 13902, USA; ehoffman@binghamton.edu; 4Department of Neurology, UCSF Benioff Children’s Hospital San Francisco, 550 16th Street, San Francisco, CA 94158, USA; jonathan.strober@ucsf.edu

**Keywords:** Duchenne muscular dystrophy, mdx, sphingosine phosphate lyase, sphingosine-1-phosphate, satellite cells

## Abstract

Duchenne muscular dystrophy (DMD) is a congenital myopathy caused by mutations in the dystrophin gene. DMD pathology is marked by myositis, muscle fiber degeneration, and eventual muscle replacement by fibrosis and adipose tissue. Satellite cells (SC) are muscle stem cells critical for muscle regeneration. Sphingosine-1-phosphate (S1P) is a bioactive sphingolipid that promotes SC proliferation, regulates lymphocyte trafficking, and is irreversibly degraded by sphingosine phosphate lyase (SPL). Here, we show that SPL is virtually absent in normal human and murine skeletal muscle but highly expressed in inflammatory infiltrates and degenerating fibers of dystrophic DMD muscle. In mdx mice that model DMD, high SPL expression is correlated with dysregulated S1P metabolism. Perinatal delivery of the SPL inhibitor LX2931 to mdx mice augmented muscle S1P and SC numbers, reduced leukocytes in peripheral blood and skeletal muscle, and attenuated muscle inflammation and degeneration. The effect on SC was also observed in SCID/mdx mice that lack mature T and B lymphocytes. Transcriptional profiling in the skeletal muscles of LX2931-treated vs. control mdx mice demonstrated changes in innate and adaptive immune functions, plasma membrane interactions with the extracellular matrix (ECM), and axon guidance, a known function of SC. Our cumulative findings suggest that by raising muscle S1P and simultaneously disrupting the chemotactic gradient required for lymphocyte egress, SPL inhibition exerts a combination of muscle-intrinsic and systemic effects that are beneficial in the context of muscular dystrophy.

## 1. Introduction

Duchenne muscular dystrophy (DMD) is a genetic condition that results in a severe myopathy marked by myositis, muscle fiber degeneration, and eventual muscle replacement by fibrosis and adipose tissue [1]. DMD is caused by mutations in the critical muscle structural protein dystrophin, resulting in progressive muscle weakness and atrophy [2]. It is the most common type of muscular dystrophy, and as an X-linked inherited neuromuscular disorder, it typically affects males. Mutations in dystrophin destabilize the dystrophin dystroglycan protein complex which normally serves a scaffolding function in muscle [3]. This leads to a degeneration of skeletal and cardiac muscle and impairs the integrity and function of satellite cells (SC) [4], the muscle stem cells critical for promoting muscle regeneration [5]. Signs and symptoms of DMD typically present during early childhood and progress steadily, with most patients becoming wheelchair-bound in their teens. Despite intense research efforts, there is still no effective cure for all boys with DMD, and corticosteroids remain the mainstay of therapy [6]. Elucidating the complex pathways that contribute to skeletal muscle degeneration, inflammation, and regeneration could lead to novel potential therapeutic strategies in DMD, including alternatives to corticosteroids [7].

Sphingosine-1-phosphate (S1P) is a bioactive sphingolipid metabolite that functions in various physiological processes, primarily by activating a family of high-affinity G protein-coupled receptors, S1P1-5 [8]. S1P signaling plays fundamental roles in lymphocyte trafficking, the maintenance of vascular integrity, inflammatory cascades, and the epigenetic regulation of gene transcription. S1P is enriched in blood and lymph while being maintained at low levels in tissues and interstitial fluids by the intracellular enzyme sphingosine phosphate lyase (SPL) [9,10]. By acting as an S1P metabolic sink, SPL activity generates the S1P chemotactic gradient that is essential for guiding the egress of mature T lymphocytes from the thymus and peripheral lymphoid organs into the blood [11]. The inhibition of SPL activity or interruption of S1P1 signaling produces a marked decrease in circulating lymphocytes due to their retention in the lymphoid organs [12]. S1P receptor antagonism has been successfully leveraged as a therapeutic strategy in certain autoimmune diseases, where it acts by preventing self-reactive lymphocytes from entering target tissues and causing inflammation and subsequent tissue damage [13].

S1P can also act as a trophic factor for skeletal muscle [14]. S1P signaling contributes to skeletal muscle homeostasis by exerting effects on muscle regeneration, atrophy, and SC functions [15]. S1P was found to attenuate dystrophic muscle phenotypes and atrophy in fruit flies and mice [16,17], whereas it rose with exercise and promoted muscle strength in non-dystrophic muscle [18]. At the cellular and molecular level, these effects have been attributed to S1P’s impact on epigenetic gene regulation, the expression of mitochondrial and metabolic genes required for muscle repair, STAT3 inflammatory signaling, TGF beta signaling, calcium mobilization, and the SC cell cycle progression, proliferation, migration to sites of injury, and ability to fuse with injured muscle fibers [16,17,19]; however, different S1P receptors play distinct and sometimes antagonistic roles, such that the ultimate impact of S1P signaling on muscle homeostasis is complex [20].

Previously, we reported that sphingolipid metabolism is dynamically regulated in response to notexin-induced muscle injury, resulting in a transient post-injury rise in circulating S1P that promotes SC proliferation and supports muscle regeneration [21]. Mdx mice that model DMD were S1P deficient, and pharmacological SPL inhibition raised the S1P levels and enhanced SC activation and muscle regeneration after notexin injury in this model. Others have also shown that a reduction in circulating S1P levels can be deleterious in mdx mice [22]. However, SPL inhibition as a strategy to raise S1P levels may not always be beneficial in the context of skeletal muscle. *Sply* mutants, which harbor a transposon insertion in the gene encoding SPL in the fruit fly, exhibit an abnormal development of thoracic flight muscles and are flightless [23]. In addition, when SPL is silenced in murine myoblasts, myogenic differentiation is compromised [24]. These findings suggest the need for more detailed studies exploring SPL targeting methods to identify the specific administration conditions that maximize the immunosuppressive and trophic effects on dystrophic muscle and SC without compromising muscle development and differentiation. In that regard, physiological models that avoid the use of toxin-mediated muscle injury would be ideal.

In this study, we examined SPL expression in the skeletal muscles of DMD patients and mdx mice. We found that SPL was upregulated in DMD and mdx muscles compared to the corresponding controls, with upregulation detected in the first few weeks of life in the mdx mice. We then investigated the impact of perinatal administration of an SPL inhibitor on various parameters in mdx mice. Our cumulative findings show that SPL inhibition early in life increased the endogenous SC pool and reduced the inflammatory component of mdx myopathy, attenuating muscle damage.

## 2. Results

### 2.1. SPL Expression Is Increased in DMD Muscles Compared to Age-Matched Controls

Quantitative real time-PCR (qRT-PCR) of frozen skeletal muscle biopsies showed a significant elevation in the expression of the SPL-encoding gene, *SGPL1*, in DMD compared to control muscles (Figure 1A). The upregulation of SPL protein in dystrophic skeletal muscle was confirmed by immunohistochemistry staining of frozen muscle tissue sections from four individuals with DMD, wherein the SPL signal was observed in infiltrating cells of the muscle interstitium as well as degenerating muscle fibers (Figure 1B). In comparison, the skeletal muscle from two age-matched controls exhibited almost no SPL staining (Figure 1B). Human DMD skeletal muscle biopsy material stained for hematoxylin and eosin (HE) showed inflammatory cell infiltrates between muscle fibers, positive staining for utrophin, but no expression of dystrophin or endothelial nitric oxide synthase, all of which are typical findings in DMD (Appendix A).

### 2.2. S1P Metabolism Is Altered in Juvenile Mdx Mice

To investigate the role of SPL in muscular dystrophy further, and with special attention to its early stages, we compared *Sgpl1* gene expression by performing qRT-PCR on the skeletal muscles (*gastrocnemius*) of mdx and wild type (WT) C57BL/10 mice at 4, 6, 8, and 10 weeks of age (Figure 2A), which represents the peak period of myositis in the mdx model. Muscle SPL expression levels trended upward from 4 to 10 weeks in both the mdx and WT pups, and this trend was more pronounced in the mdx muscle, which exhibited higher SPL levels than their WT counterparts at all ages (Figure 2A). The changes in muscle *SGPL1* expression over time were accompanied by a steady decline in skeletal muscle S1P, observed in both *diaphragm* and *tibialis anterior* muscles over the six-week period in both groups, with more pronounced effects observed in the mdx group (Figure 2B,C). Furthermore, the plasma S1P levels were markedly reduced in the mdx mice compared to the WT mice at four and six weeks of life (Figure 2D). By immunofluorescence microscopy, SPL staining was prominently detected in mdx muscles (*tibialis anterior*) at all ages, whereas SPL staining in the corresponding WT muscles was absent (Figure 2E). Tissue sections co-stained with anti-CD45 to identify leukocytes revealed that most of the SPL signal in mdx muscle was found in CD45-positive cells located within the inflammatory infiltrates and necrotic fibers (Figure 2E and Appendix A). These findings confirm a previous report identifying *Sgpl1* as a component of an inflammatory gene signature in dystrophic mdx muscle [25]. Additionally, our findings that both DMD and mdx skeletal muscle show a similar upregulation of SPL gene and protein expression, confirm that the mdx mouse model is appropriate for further investigation of the role of SPL in muscular dystrophy.

### 2.3. Perinatal Treatment of Mdx Mice with SPL Inhibitor Reduces Immune Cell Trafficking to Skeletal Muscles

We hypothesized that raising the blood and tissue S1P levels by inhibiting its metabolism at a critical period before dystrophy begins to manifest in mdx pups might attenuate their dystrophic features. Since we observed *Sgpl1* upregulation in the skeletal muscles of mdx pups by week four of life, a perinatal delivery method was devised to deliver the SPL inhibitor LX2931 to the mothers throughout pregnancy and nursing, followed by the analysis of their pups. The perinatal delivery method consisted of dissolving the SPL inhibitor in drinking water and administering it *ad libitum* to mdx mothers during pregnancy and nursing, followed by analysis of their weaning pups at four weeks of age. We confirmed the effect of SPL inhibition by detecting leukopenia and lymphopenia in the white blood cell counts of perinatally treated mdx pups as compared to the untreated mdx controls (Figure 3A,B). The S1P levels in diaphragm skeletal muscle were augmented in the LX2931-treated pups as compared to the controls (Figure 3C); however, the plasma S1P levels were not different between the control and treated pups (Figure 3D). Immune profiling showed that CD4+ and CD8+ lymphocytes in the blood of the LX2931-treated mdx pups were reduced compared to the untreated mdx pups (Figure 3E). In addition, we observed a reduction in circulating neutrophils and monocytes in the pups treated with LX2931 (Figure 3E). The immune profile of *gastrocnemius* muscle isolated from the LX2931-treated mdx pups showed reduced CD4+ lymphocytes, B cells and neutrophils compared to untreated control mdx muscle (Figure 3F); thus, SPL inhibition reduced muscle tissue infiltration by cells of both the innate and adaptive immune systems.

Leukocytic infiltration was next compared in the *tibialis anterior* muscles of LX2931-treated mdx and untreated mdx pups using immunofluorescence detection of CD45+ cells in cryosections.

A marked reduction in inflammatory infiltrates was observed in the treatment group, as shown in Figure 4A, with the signal quantified in Figure 4B. The predominant leukocyte cell population present in the muscles of pups were neutrophils (Figure 3F). The identification of neutrophils by immunofluorescence detection of the neutrophil marker, elastase, confirmed a reduction in the neutrophil infiltrates in the muscles of the LX2931-treated mice compared to the untreated controls, as shown in Figure 4C and signal quantified in Figure 4D. These findings demonstrate that SPL inhibition reduces inflammation in dystrophic muscles.

### 2.4. Perinatal SPL Inhibition in Mdx Mice Attenuates Muscle Damage

To assess the impact of SPL inhibition on the dystrophic muscle injury incurred by normal muscle contraction during daily activity, we measured the number of regenerating fibers by counting the nucleated fibers in the fixed skeletal muscle sections of LX2931-treated and untreated mdx mice. At three weeks of age, the LX2931-treated pups exhibited profoundly reduced muscle damage compared to the untreated mdx pups, shown by the lower number of regenerating myofibers with characteristic centrally located nuclei and the lower number of inflammatory cell infiltrates in the *tibialis anterior* muscles (Figure 5A–C). These findings support the notion that SPL inhibition delays the onset of muscle damage and/or prevents the muscle damage characteristic of mdx mice at this age.

### 2.5. Perinatal SPL Inhibition Increases Endogenous Satellite Cell (SC) Numbers in Mdx Muscles

When the SC numbers in fresh *gastrocnemius* skeletal muscles of pups treated on the LX2931 perinatal regimen were compared to those of the control pups at three weeks of age, they were found to be significantly elevated in the LX2931-treated mice, as determined by flow cytometry (Figure 6A). Consistent with this finding, the gene expression levels of the SC markers, *Pax7* and *Met*, were also higher in the LX2931-treated versus the untreated pups, as measured by qRT-PCR (Figure 6B). These cumulative findings confirm that SPL inhibition early in life promotes endogenous SC expansion.

### 2.6. Perinatal SPL Inhibition Diminishes Fibrosis in Mdx Muscles

The normal progression of DMD involves the replacement of muscle fibers by fibrosis and, eventually, by adipose tissue. In mdx mice, florid muscle fibrosis develops in the adult phase; however, the early onset of collagen deposition can be observed [26]. Cryosections of diaphragm skeletal muscle harvested from three-week-old LX2931-treated and untreated control mdx pups were stained with Masson’s trichrome to detect collagen. The control muscles exhibited more collagen deposition than muscles from the LX2931-treated pups, as shown in Figure 7A and quantified in Figure 7B. To further evaluate the effect of the LX2931 treatment on fibrosis, the expression of four representative fibrosis genes, *Col1a1*, *Col3a1*, *Col61*, and *Fn1*, encoding collagens and fibronectin were compared by qRT-PCR in the frozen diaphragm muscle tissues from both groups. The expression levels of *Col-6a1* and *Fn1* were significantly lower in the muscles of LX2931-treated pups compared to the untreated mdx control group (Figure 7C). *Col-1a1* expression trended lower in the treated pups but was not significantly reduced, and the *Col-3a1* levels were not different in the two groups (data not shown). These findings suggest that perinatal SPL inhibition may reduce the early deposition of collagen in mdx muscles.

### 2.7. SPL Suppression Results in Increased Muscle Strength

To determine the functional significance of the reduced inflammation, reduced injury, and increased SC numbers in LX2931-treated mice, muscle strength was measured in six-week-old LX2931-treated and control mice. Importantly, grip strength normalized to the body weight was significantly improved in the LX2931-treated mice, indicating that SPL inhibition may lead to a functional improvement in dystrophic muscles (Figure 8).

### 2.8. SPL Inhibition Promotes SC Expansion in Immunodeficient Mdx Mice

Inhibition of SPL and S1P1 antagonism both block T lymphocyte egress, causing the depletion of circulating T cells; however, unlike the S1P1 antagonists, SPL inhibition also reduces infiltrating neutrophils and monocytes. To delineate the contribution of the lymphocyte-specific effects of SPL inhibition on SC expansion in mdx mice, LX2931 was administered using the perinatal delivery method to mdx mice and to SCID/mdx mice, which lack T and B cells. The SC abundance was then compared by flow cytometry of the *gastrocnemius* muscle tissues of LX2931-treated and control mice of both groups. As shown in Figure 9, SPL inhibition significantly increased the SC numbers in the mdx mice of both genetic backgrounds, although SC expansion in response to LX2931 was less robust in the SCID/mdx group compared to the mdx group. These findings suggest that both lymphocyte-specific and lymphocyte-independent effects may contribute to SC expansion in response to SPL inhibition.

### 2.9. Transcriptional Profiling in LX2931-Treated and Untreated Mdx Muscle

A comparative global gene expression analysis was performed on the skeletal muscles (*tibialis anterior*) of the LX2931-treated and untreated mdx mice (*n* = 3/group, each sample performed in duplicate) harvested at four weeks of life. A principal component analysis (PCA) was performed to evaluate any differences among the biological replicates and their treatment conditions. The variable of the first three principal components (PC1, PC2 and PC3) were 83.4, 12.10 and 2.20%, respectively. The PCA revealed a homogeneous expression pattern for the LX2931-treated group, and a less cohesive distribution for the untreated group, but a clear separation between the treated and untreated *tibialis anterior* muscle gene expression patterns (Figure 10A). A clustering analysis was performed to visualize the correlations among the replicates and the varying sample conditions. This revealed 285 genes differentially up- and down-regulated in the two groups (Figure 10B). The most enriched pathways of the differentially expressed genes were the leukocyte trans-endothelial migration, ECM-receptor interactions, and chemokine signaling (Table 1). The most enriched molecular functions of the differentially expressed genes included calcium, carbohydrate, polysaccharide, protein, protein complex, and pattern binding, as shown in Table 2. In Table 3, the biological processes most affected by perinatal SPL inhibition include the functions related to immunity, leukocyte biology, cell adhesion, and activation, wound healing, and vascular development. Taken as a whole, these findings are consistent with the observed reduction in infiltrating leukocytes in the LX2931-treated mdx muscle as well as the known roles of S1P in lymphocyte trafficking, angiogenesis, and calcium homeostasis. Table 4 shows that SPL inhibition overwhelmingly affected the cell surface and plasma membrane-related proteins, ECM, and collagen, likely reflecting the impact on the sarcolemma and its interactions with the ECM. A pathway analysis showed a differential impact on the key pathways of immune function and cell signaling including leukocyte trans-endothelial migration, ECM-receptor interactions, chemokine signaling, Fc gamma receptor-mediated phagocytosis, hematopoietic cell lineage, cell adhesion molecules, B cell receptor signaling, natural killer cell mediated cytotoxicity, focal adhesion, and axon guidance (Appendix A). Most of these pathways were globally downregulated in the LX2931-treated muscle, reflecting the profound impact of SPL inhibition on circulating and tissue infiltrating immune cells.

## 3. Discussion

Although DMD is a monogenic disease, the underlying defect in dystrophin destabilizes muscle-ECM interactions, stimulating a complex array of intracellular and extracellular consequences. These events cause muscle breakdown, myositis, and compromised stem cell functions, setting off an ineffective chronic wound healing response that ultimately leads to the replacement of muscle by adipose and scar tissue. An improved understanding of the cascade of the molecular events involved in DMD may reveal novel therapeutic strategies. In this study, we found that SPL, a key enzyme of sphingolipid metabolism, is upregulated in the muscles of DMD patients and mdx mice. The gradual increase in muscle SPL expression in juvenile mice appears to be a physiologic event, as it is observed even in WT mice; however, the trend is exaggerated in the dystrophic muscles of mdx mice, where SPL was found primarily in the inflammatory cells and necrotic muscle fibers, accompanied by the dysregulation of S1P metabolism. These findings are consistent with our previous results showing that muscle injury provokes dynamic changes in muscle S1P metabolism [21]. Based on this observation, we reasoned that boosting S1P through SPL inhibition early in life might promote positive effects on skeletal muscle through the various actions of S1P signaling, thereby attenuating dystrophic changes.

To test this hypothesis, we developed a noninvasive method of inhibiting SPL early in life by the oral *ad libitum* administration of LX2931 to pregnant and nursing mdx mothers. The treatment was well tolerated by the mothers, who maintained healthy body condition scores throughout pregnancy and nursing, yielded the expected litter size, nursed effectively, and were able to have subsequent pregnancies (data not shown). The pups also tolerated the treatment well, as shown by the lack of significant differences in weight gain from the untreated mdx controls, and no indication of infection or any other signs of poor health (data not shown). As expected, pups of the mothers treated with LX2931 exhibited the hallmarks of SPL inhibition, namely, tissue S1P elevation and T cell lymphopenia resulting from blocked lymphocyte egress. In addition, flow cytometry revealed a reduction in the circulating B cells, neutrophils, and monocytes as well as a reduced infiltration of mdx muscle with CD4+ T cells, B cells, and neutrophils. These findings were further confirmed by immunofluorescence microscopy. The reduced emigration of neutrophils into the tissues was consistent with a previous report showing that SPL disruption interferes with neutrophil entry into tissues via an S1P4-dependent process [27].

Perinatal SPL inhibition attenuated the dystrophic muscle changes as evidenced by fewer regenerating muscle fibers, reduced inflammation, and improved grip strength. While we also found a modest impact of SPL inhibition on the levels of fibrosis in the diaphragms of treated compared to untreated mdx mice, these results should be interpreted with some degree of caution. Our finding of 7% fibrosis in untreated mdx diaphragm muscle was considerably lower than the 11% reported by Gutpell et al. [28], and the reduction to 2% fibrosis in our treatment group was lower than the 7% fibrosis in the WT diaphragm reported in the same study. The discordance of fibrosis measurements between our study and the Gutpell report could be due to differences in our staining technique, or epigenetic changes or environmental conditions in our mdx colony and/or animal facility, respectively.

A limitation of our study is that we utilized three different muscle groups throughout our study, including the *tibialis anterior*, *gastrocnemius* and diaphragm. This decision was made so as to obtain as much information as possible from the small amount of muscle tissue available from juvenile mice. In all experiments, we compared endpoints in the same muscle type in both the treatment and control subjects; however, these muscle groups are different in many respects, and it would be ideal in future studies to compare all relevant endpoints in all three muscle groups to look for differences that may exist. Our initial comparison of mdx and WT control mice revealed alterations in sphingolipid metabolism including the upregulation of *Sgp1l* and reduction in its substrate in mdx mice. We did not include a WT cohort in subsequent experiments, as our focus was to interrogate the impact of SPL inhibition on the features of mdx mice; however, future studies may benefit from the inclusion of a control cohort for comparison. One additional limitation of our study is that we did not follow the long-term outcome of the treated mice, which were instead sacrificed before reaching maturity for histological and molecular analysis. In the future, studies conducted over longer time periods will be required to establish the true potential of SPL inhibition as a strategy to mitigate muscular dystrophy. This is particularly important in consideration of the upregulated inflammatory cytokines observed in global *Sgpl1* knockout mice [27,29]. Encouragingly, our transcriptional profiling showed a suppression of STAT activation and chemokine signaling and no obvious upregulation of cytokines in LX2931-treated subjects. Further, LX2931 affords a partial rather than complete inhibition of SPL and may not produce the same effects as complete SPL inhibition. Future studies should nonetheless pay close attention to the possible unwanted effects of SPL inhibition.

Importantly, SPL inhibition also led to the expansion of the SC population, as shown by the direct measurement by flow cytometry, complemented by a quantification of the muscle expression of the SC markers, *Pax7* and *Met*. Expansion of the SC pool was also observed (albeit to a lesser degree) in the SCID/mdx mice. This suggests that perinatal SPL inhibition promotes SC expansion independently of its impact on lymphocyte trafficking. The effect on SCs may be attributed to a direct trophic effect of S1P signaling on the SC population. Notably, S1P receptor antagonists inhibit lymphocyte trafficking by downregulating S1P receptors; however, they do not appreciably alter S1P levels. In contrast, SPL inhibition blocks lymphocyte trafficking by inducing S1P accumulation in the tissues, thereby disrupting the S1P chemotactic gradient. Thus, additional trophic effects of S1P and the effects on the innate immune system mediated by SPL inhibition may be leveraged to prevent inflammation and mitigate dystrophy in a way that S1P receptor antagonists do not.

We undertook transcriptional profiling in the *tibialis anterior* muscles of LX2931-treated vs. control mdx mice to further delineate the impact of SPL inhibition on mdx muscle. Transcriptional changes in the muscles of young mdx mice compared to controls have been well-documented, with many studies revealing a consistent pattern of hundreds of differentially expressed genes in mdx muscle, the overwhelming majority of these being upregulated transcripts involved in the immune response, phagocytosis, ECM, and wound healing/regeneration processes [30,31,32,33,34]; thus, even though the mutation of a single gene (dystrophin) initiates the DMD myopathy, a complex immunological process ensues that contributes to the disease progression. We found that SPL inhibition reversed this process at least in part, with the majority of differentially expressed transcripts in the LX2931-treated group being downregulated genes with immune-related functions. Specifically, we observed global changes in innate and adaptive immunity, plasma membrane interactions with the ECM, and axon guidance. SPL inhibition reduced the expression of genes associated with leukocyte invasion, phagocytosis, and B and NK cell activation. Reduced levels of circulating T, B, and NK cells, and monocytes have been well documented in children with SPL insufficiency syndrome, an inborn error of metabolism caused by mutations in the human SPL gene, *SGPL1* [5]. In the case of DMD, inhibiting phagocyte activation and migration into tissues could be particularly beneficial in avoiding further damage to muscles induced by inflammation. The finding of a major influence on ECM-receptor interactions may be indicative of the stabilizing function of the dystroglycan-associated protein complex located in the sarcolemma, which is anchored to the cytoskeleton intracellularly and the ECM extracellularly. The absence of full-length dystrophin in mdx muscle partially compromises this interaction, which may be protected by SPL inhibition. The effects of SPL inhibition on genes associated with axon guidance could potentially be related to changes in the SC population, as SCs are known to produce and export axon guidance molecules in response to muscle injury [35,36,37,38]. This finding seems to be unique to our study and is particularly interesting in consideration of the peripheral neuropathy and axonopathy associated with *SGPL1* mutations and SPL insufficiency syndrome [39]. Further elucidation of the role of SPL in axonal regeneration and axon guidance cues may be important future areas of study.

In the last few years, substantial progress has been made leveraging gene therapy, gene editing, and exon skipping to address the root cause of DMD. There is much hope that with continued technological advances these approaches will eventually improve the clinical outcomes for the majority of DMD patients; however, only a limited number of patients have benefitted to this day. In the near term, a meaningful and attainable goal would be the identification of alternatives to steroids, the current mainstays of DMD treatment, which suppress inflammation but are not curative and have profound unwanted side effects [40].

Our cumulative findings suggest that SPL inhibition should be further explored as a potential treatment of DMD. This strategy provides a potentially safer and more tolerable alternative to steroids as a method of immune suppression to prevent myositis. At the same time, SPL inhibition augments tissue levels of the trophic muscle factor S1P, leading to an expansion of the SC population, thereby potentially improving the regenerative capacity of dystrophic muscle. Multiple SPL inhibitors have been generated to date [41,42]. LX2931 has the advantage of having been tested in human clinical trials [43]. Our novel method of delivering LX2931 perinatally was well tolerated by both mdx pups and their mothers. Our findings suggest that when the diagnosis of DMD is known early in life or potentially even prenatally, prompt intervention might avert the inflammatory component of the disease, delaying progression until more definitive therapy such as gene therapy/editing or antisense oligonucleotides can be administered. Future studies will be required to confirm the utility of this strategy.

## 4. Materials and Methods

### 4.1. Human Tissues

Skeletal muscle biopsies from DMD patients and controls used in the qRT-PCR experiments were obtained with informed consent at the National Children’s Medical Center by EPH. Deidentified skeletal muscle biopsies (*gastrocnemius* or *tibialis anterior*) of the DMD patients and controls used for the immunohistochemistry study were stained and imaged by the Wellstone Muscular Dystrophy Research Center at the University of Iowa. The polyclonal goat antibody against SPL was from R&D Systems.

### 4.2. Mice

WT (C57BL/10ScSnJ) mice were obtained from the Jackson Laboratory (Bar Harbor, Maine). Mdx breeders (C57BL/10ScSn-*Dmd^mdx^*/J) and SCID/mdx (B10ScSn.Cg-Prkdc*^scid^* Dmd*^mdx^*/J) mice were purchased from the Jackson Laboratory. The mice were housed in a temperature- and humidity-controlled room and maintained on a light–dark cycle with food and water *ad libitum*. The mice were acclimated to the animal facility for a minimum of two weeks prior to enrollment in the study.

### 4.3. LX2931 Administration

The oral administration of 500 mg/L of LX2931 (provided by Lexicon Pharmaceuticals, The Woodlands, TX, USA) was given to the mice *ad libitum* in the drinking water and was refreshed every 2–3 days [44]. The control mice received normal drinking water. For the immunodeficient mice, water was autoclaved and prepared in sterile conditions.

### 4.4. Tissue Harvesting

Unless otherwise stated, the pups of mothers treated with LX2931, and the controls were euthanized at 21 days and their tissues collected. Blood was collected by cardiac puncture and processed for flow cytometry or lipid analysis. The skeletal muscles (*gastrocnemius*, *tibialis anterior* and diaphragm) were excised and either processed for flow cytometry or snap-frozen in liquid nitrogen for RNA extraction and the measurement of S1P levels. The muscle groups used for each experimental endpoint are indicated throughout the text. Some of the collected muscles were cryopreserved after mounting the tissue on a base of cork with tragacanth (Sigma-Aldrich, St. Louis, MO, USA), by submersion in liquid nitrogen-cooled 2-methylbutane (Fisher Scientific, Hampton, NH, USA) for 1 min and then stored at −80 °C for immunohistochemistry.

### 4.5. Flow Cytometry

For analysis via flow cytometry, the *gastrocnemius* muscles were excised, minced, and digested in Dulbecco’s modified Eagle’s medium (DMEM, Grand Island, NY, USA; Invitrogen, Waltham, MA, USA) containing 0.1% of collagenase type 2 (315 U/mg, CLS2, Worthington, Lakewood, NJ, USA) and 100 U/mL penicillin G, and 100 µg/mL streptomycin (P/S; University of California San Francisco, UCSF, CA, USA) for 30 min at 37 °C while shaking. To stop the enzymatic digestion, an equal volume of DMEM containing P/S and 10% fetal bovine serum (Sigma-Aldrich) was added for 30 min at 37 °C while shaking. The digested product was filtered through a 70 µm cell strainer and washed with phosphate-buffered saline (PBS). The filtrated suspension was centrifuged at 1200 RPM for 8 min. After removing the supernatant, the cell pellet was resuspended in 5 mL PBS containing 0.1% bovine serum albumin (BSA) to count the cells (4–12 µm size) and for antibody labeling. Two million cells of each sample were stained with the following conjugated anti-mouse antibodies for leukocytes: CD4-eFLuor 450 (clone RM 4-5), CD8a-FITC (clone 53–3.7), CD11b-PE (clone M1/70), F4/80-PE Cy7 (clone BM8), Ly-6G (Gr1)-PerCP Cy5.5 (clone RB6–8C5) and CD45R (B220)-APC (clone RA3–6B2), all from eBioscience, Inc. (San Diego, CA, USA). One million cells were stained for myogenic cells (SC) using a cocktail containing the following conjugated anti-mouse antibodies: Integrin α7-APC (rat IgG2b; R&D, Minneapolis, MN, USA), CD34-eFluor 450 (clone RAM34), Ly-6A/E (Sca1)-PerCP Cy5.5 (clone D7), CD31-PE Cy7 (clone 390), all from eBioscience, San Diego, CA, USA and CD45-FITC (Bio-Rad, Hercules, CA, USA). The blood was collected in EDTA collecting tubes. To remove the red blood cells, the blood was processed in three cycles of incubation for 5 min at RT with an AKC lysing buffer (1.5 ammonium chloride, 100 mM potassium bicarbonate and 100mM EDTA). Cold PBS was added to deactivate the AKC buffer, and the sample was centrifuged for 5 min at 1200 RPM at RT. In the last cycle, the cell pellet was resuspended in 5 mL PBS containing 0.2% BSA and filtered using 40 µm cell strainers. A cell count was performed, and two million cells were incubated with the six antibodies used for leukocyte detection described above. The signals were acquired with a FACS Fortessa (BD Bioscience, San Jose, CA, USA), and the data were analyzed with FlowJo, Ashland, OH, USA (Tree star).

### 4.6. S1P Quantification

The tissues were homogenized in 0.25 mL methanol (HPLC grade, Fisher Scientific) using a tip sonicator (60 Sonic Dismembrator, Fisher Scientific, Pittsburgh, PA, USA). A homogenate was combined with C17-S1P (Avanti Polar Lipids, Alabaster, AL, USA) internal standard and 0.5 mL of chloroform (HPLC grade, Fisher Scientific):methanol (1:1), and the sample was incubated overnight at 48 °C. The samples were dried down with nitrogen gas and resuspended in 0.5 mL of chloroform:methanol (2:1). From this suspension, 50 µL were removed and dried down with nitrogen. The extract was made basic by adding 50 µL of 1M KOH in methanol. To accomplish the two-phase separation, 100 µL of water was added. The aqueous phase was transferred to a new tube and made acidic by adding 35 µL of acetic acid. A second aqueous phase was created by adding 0.2 mL of chloroform:methanol (2:1) and 50 µL of water. The organic phase was recovered, dried down with nitrogen, resuspended in 50 µL of methanol containing 5 mM of ammonium acetate, mixed by vortexing, and placed in a water bath sonication device for 5 min before a S1P measurement by mass spectrometry. 17C-S1P (Avanti Polar Lipids, Alabaster, AL, USA) was used as an internal standard. The lipids were separated on a C18 column (2.1 × 50 mm; Kinetex, Phenomenex, Torrance, CA, USA) at a flow rate of 0.3 mL/min. The data were acquired on a Micromass Quattro LCZ (Waters Corp., Milford, MA, USA) mass spectrometer and processed by MassLynx v3.3. The lipids were identified based on their specific precursor and product ion pair and quantitated using multiple reactions monitoring as described [45]. The lipid normalization was accomplished by a phospholipid determination (Phospholipids C, Wako Diagnostics, Richmond, VI, USA).

### 4.7. qRT-PCR

The RNA from skeletal muscle was extracted using an RNeasy mini kit (Qiagen) following the manufacturer’s protocol. The RNA was eluted in a 30 µL elution buffer, and the concentration was determined with a NanoDrop ND 1000 spectrophotometer. The cDNA was generated from 1 µg of RNA using the First-Strand cDNA Synthesis kit (Invitrogen-Life Technologies, Carlsbad, CA, USA) run on a PTC-240 Tetrad 2 DNA machine (Bio-Rad, Hercules, CA, USA) using the cycling conditions recommended by the manufacturer. The qPCR was performed using a Power Sybr green PCR master mix (AB Life Technologies). The cycling parameters used for all primers were as follows: 95 °C for 10 min; PCR 40 cycles of: 30 s. 95 °C denaturation, 1 min. 55–60 °C annealing, 30 sec. 72 °C extension, and were repeated. A DNA dissociation curve was performed for each sample, to ensure the purity of the amplification products. The amplification of GAPDH was utilized as the internal control to normalize the data. The primers used in this study (Table 5) were obtained from Integrated DNA Technologies (IDT, Coralville, IA, USA). Each sample was run in triplicate, and the average of the three was used for further calculations.

### 4.8. Microarray Analysis

The total RNA was isolated from the *tibialis anterior* muscles of perinatally LX2931-treated mice (*n* = 3) and control mice (*n* = 3) at 4 weeks of age using a Qiagen RNeasy mini kit according to the manufacturer’s protocol. The RNA samples were treated with DNA digestion to remove genomic DNA contamination. The total RNA concentration and purity were assessed by measuring the optical density (230, 260, and 280 nm) with the Nanodrop 1000 Spectrophotometer (ThermoFisher Scientific, Waltham, MA, USA). The RNA samples were sent to Phalanx Biotech (San Diego, CA, USA). The Fluorescent aRNA targets were prepared from 1 μg of the total RNA samples using a OneArray^®^ Amino Allyl aRNA Amplification Kit (Phalanx Biotech Group, Hsinchu, Taiwan) and Cy5 dye (GE Healthcare, Chicago, IL, USA). Fluorescent targets were hybridized to the Mouse Whole Genome OneArray^®^ with a Phalanx hybridization buffer using the Phalanx Hybridization System. After 16 hr hybridization, non-specific binding targets were washed away. The slides were scanned using a DNA Microarray Scanner (Model G2505C, Agilent Technologies, Santa Clara, CA, USA). The Cy5 fluorescent intensities of each spot were analyzed using the GenePix 4.1 software (Molecular Devices). Each single sample was performed in duplicate, and replicates provided reproducibility ≥0.975. The signal intensity was loaded into the Rosetta Resolver System^®^ (Rosetta Biosoftware, Seattle, WA, USA) for data preprocessing.

### 4.9. Histology

Cryopreserved muscle tissue sections (6 µm) were prepared on SuperFrost Plus glass slides (Fisher) and stored at −80 °C. Centralized nucleated myofibers were quantified after hematoxylin, eosin (Sigma-Aldrich) and saffron (HES) staining. The collagen deposits were evaluated by Masson’s trichrome staining using a kit (American MasterTech, Lodi, CA, USA) according to the manufacturer’s protocol. Six representative sections imaged at 200× magnification were analyzed from each muscle. The images were acquired by Camera QImaging (Retiga EX, Canada) mounted in a Nikon Eclipse E600 microscope (Tokyo, Japan) with iVision software (version v 4.0.11, Biovision Tech, Exton, PA, USA). The quantitation of the results was performed using Image J software (version 1.46r; National Institutes of Health, Bethesda, MD, USA).

### 4.10. Immunofluorescence

The frozen tissue sections were thawed and fixed with cold acetone for 15 min, followed by two PBS rinses of 5 min each. To block the endogenous Fc receptor binding sites, the sections were incubated for 1 hr with 10% normal goat serum (Vector Laboratories, Burlingame, CA, USA). The sections were then labeled with a blocking buffer containing primary antibodies or appropriate isotype controls overnight at 4 °C. Successive washes in PBS were followed by another 60 min incubation with the appropriate secondary antibodies at room temperature. The sections were then washed with PBS and nuclei labeled with 4′,6-diamidino-2-phenylindole (DAPI, Sigma-Aldrich) in PBS. The primary antibodies included rabbit anti-SPL (Genemed Synthesys, San Antonio, TX, USA), rat anti-CD45 (clone I3/2.3; Santa Cruz Biotechnology, Dallas, TX, USA) for the murine tissues, mouse anti-CD45 (clone 2D-1; Santa Cruz Biotechnology) for the human tissues, developmental myosin heavy chain (MHC; Vector Laboratories), and rat anti-F4/80 (BioRad), applied at 1:100. The secondary antibodies were AlexaFluor 488 or 546 specific to various species (Invitrogen, 1:400). For the mouse anti-Pax7 (Development Studies Hybridoma Bank, Iowa City, Iowa) and rabbit anti-laminin (Sigma-Aldrich), the tissue sections were immersed twice in a 10 mM citric acid solution (pH 6.0) for 5 min at 90 °C. Following the cooling and washing steps, the sections were blocked for 2 to 3 h with 4% IgG- and protease-free bovine serum albumin (Jackson ImmunoReseach, West Grove, PA, USA) in PBS. Next, the sections were sequentially incubated with goat anti-mouse IgG (H + L) AffiniPure Fab fragment (MouseFab, Jackson ImmunoResearch; dilution 0.05 mg/mL) for 30 min, with the anti-Pax7 antibody overnight, biotin-SP-conjugated AffiniPure goat anti-mouse IgG (H+L) (Jackson ImmunoResearch; dilution 1:20) for 45 min, and Cy3-conjuated streptavidin (Jackson ImmunoResearch; dilution 1:1250) for 30 min. Each incubation was followed by three rinses with PBS. Next, the sections were stained for 10 min at room temperature with DAPI in PBS, washed thrice with PBS to remove the excess dye, and mounted in Vectashield mounting medium (Vector Laboratories). The microscopic analysis was performed with a LSM T-PMT, Zeiss microscope. The images were captured with an Axiocam MRc5 camera and archived using Zen microscope software. Photoshop CS4 (Adobe, San Jose, CA, USA) was used to globally process all images for contrast, size, and brightness. Images were analyzed using ImageJ software to detect each stain. Results are given in pixel area and then converted to percentage & averaged, with 100% representing the total 324 x 324 pixel area of the image.

### 4.11. Functional Test

Grip force on the forelimbs was measured by a grip strength meter (Columbus Instruments, Columbus, OH, USA) and normalized to body weight for the assessment of functional performance in mdx mice, as described previously [46].

### 4.12. Statistics

Values are expressed as means ± standard deviation (SD) or standard error of the mean (SEM). A Student’s unpaired *t*-test was used to compare two data sets. A Tukey’s multiple comparison test was applied when comparing more than two data sets. A statistical significance was considered *p* ≤ 0.05. All experiments were conducted at least three separate times, and the results shown are representative results.

## Figures and Tables

**Figure 1 ijms-23-07579-f001:**
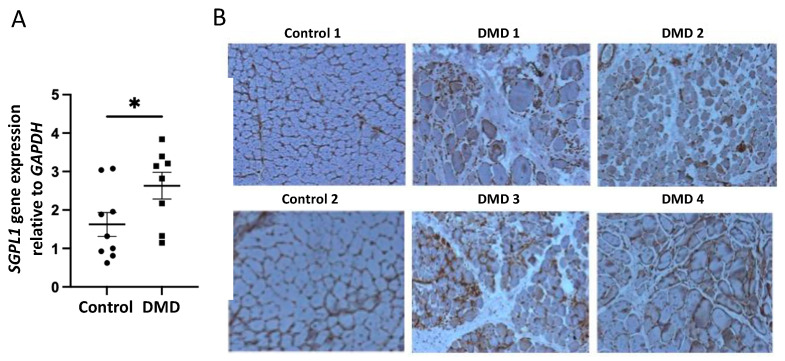
*SGPL1*/SPL is upregulated in dystrophic muscles of DMD patients. (**A**) qRT-PCR of *SGPL1* human gene expression in DMD skeletal muscle (*n* = 8) compared to control muscle (*n* = 9). Gene expression is normalized to *GAPDH*. Mean ± SD; * *p* < 0.05. (**B**) SPL protein expression as shown by immunohistochemistry (brown signal) in two controls and four DMD patients. Controls and DMD-1, -2 and -3 are from *gastrocnemius*. DMD-4 is from *tibialis anterior*.

**Figure 2 ijms-23-07579-f002:**
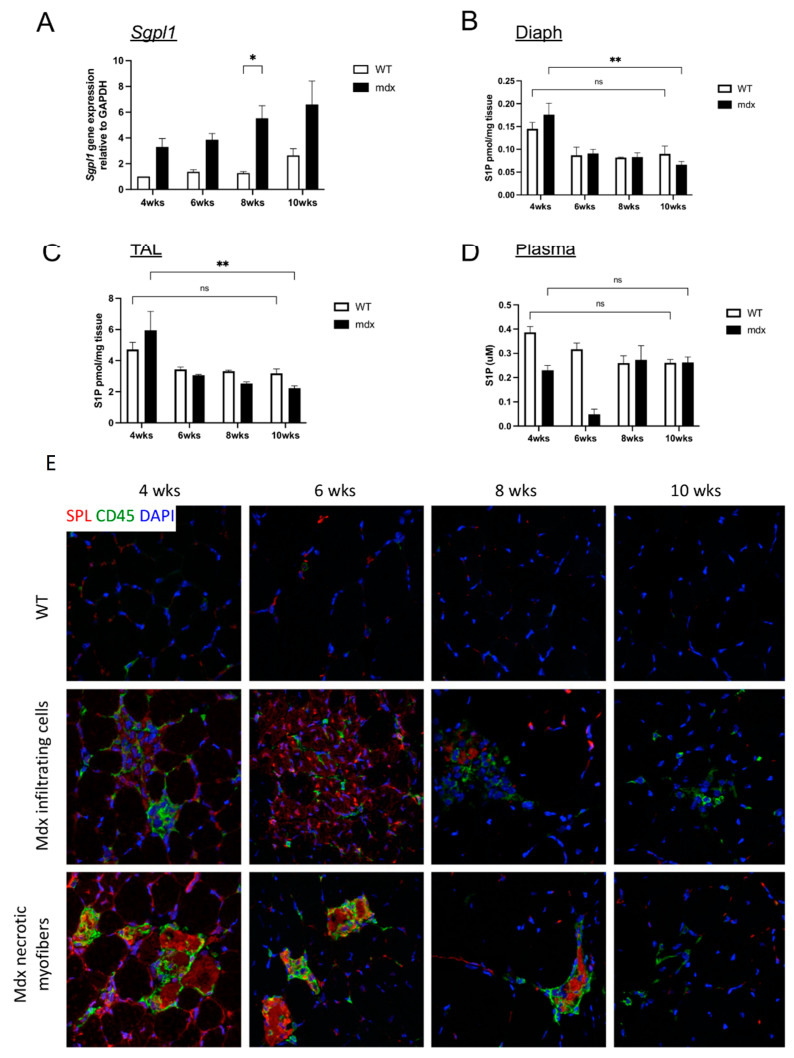
*Sgpl1*/SPL is upregulated in dystrophic muscles of mdx mice. (**A**) Expression of the murine *Sgpl1* gene in *gastrocnemius* muscles of mdx mice (black bars) and wild type C57BL/10 control mice (WT, white bars), measured by qRT-PCR. Gene expression is normalized to *Gapdh*. Mean ± SD, *n* = 3/group. S1P levels in skeletal muscles; diaphragm (**B**); left *tibialis anterior* (**C**) of WT and mdx mice measured by liquid chromatography/mass spectrometry and normalized to wet weight. (**D**) S1P levels in plasma of WT and mdx mice measured by liquid chromatography/mass spectrometry. Mean ± SD. *n* = 3/group; * *p* < 0.05, ** *p* < 0.01. (**E**) Immunofluorescence detection of SPL in skeletal muscle (*tibialis anterior*) cryosections from WT (upper panels) and mdx (middle and bottom panels) mice of different ages illustrates the SPL signal in the inflammatory cell infiltrates. For all panels, green = CD45 staining of leukocytes, red = SPL staining, blue = DAPI staining of nuclei. Magnification = 400×.

**Figure 3 ijms-23-07579-f003:**
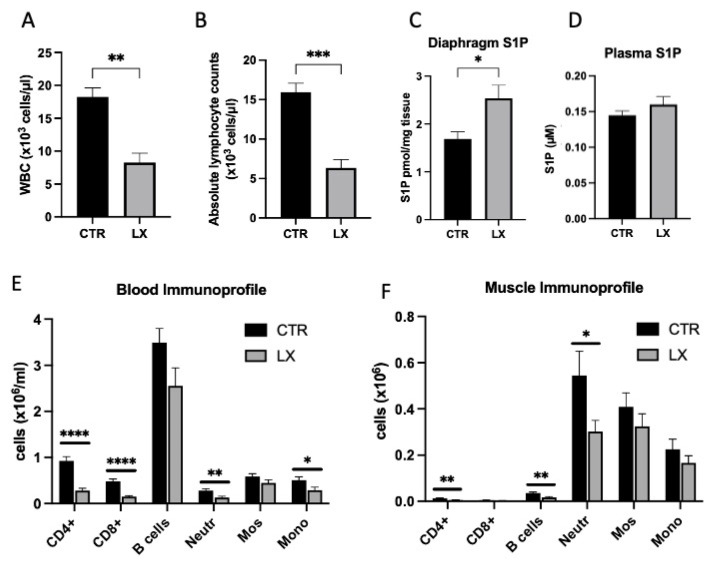
LX2931 perinatal treatment raises skeletal muscle S1P levels and induces lymphopenia in mdx mice. LX2931 (LX) treatment of female mdx mice throughout pregnancy and nursing results in leukopenia (**A**) and lymphopenia (**B**) in their 4-week-old pups compared to controls (CTR). (**C**) Diaphragm S1P levels are elevated in pups treated with perinatal LX2931. (**D**) Plasma S1P levels are unchanged by perinatal LX2931 treatment. (**E**) Flow cytometry analysis of leukocyte subsets in whole blood of control and LX2931-treated mdx pups. One million white cells were incubated with a cocktail of antibodies specific for CD4+ T cells, CD8+ T cells, B cells, neutrophils (Neutr), monocytes (Mono) and F4/80+ monocytes (Mos). One-hundred thousand events were scored. Results were analyzed using FlowJo software. (**F**) Flow cytometry analysis of the same leukocyte subsets present in the skeletal muscles (*gastrocnemius*) of control and LX2931-treated mdx pups. Two-hundred thousand events were scored. Results were normalized to dissociated muscle cell number. For (**A**–**F**), results shown are mean ± SD; * *p* < 0.05, ** *p* < 0.01, *** *p* < 0.005, **** *p* < 0.001; *n* = 3–5/group.

**Figure 4 ijms-23-07579-f004:**
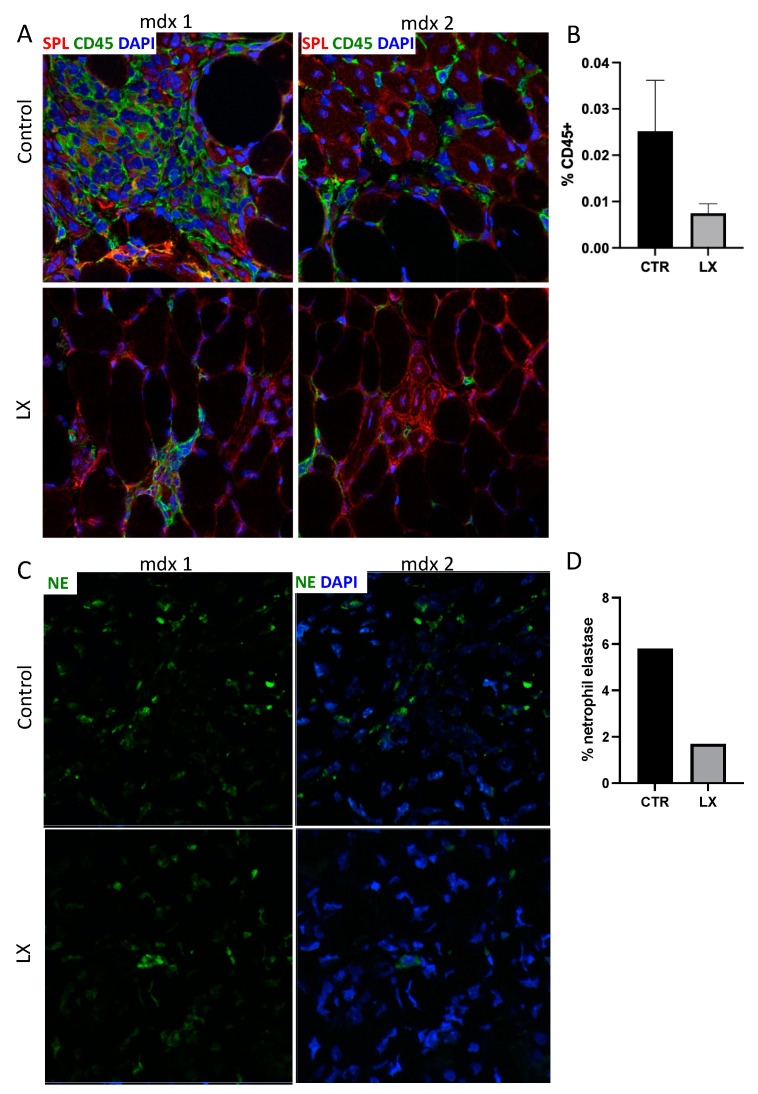
LX2931 perinatal treatment reduces skeletal muscle inflammation in mdx mice. (**A**) Immunofluorescence microscopy detecting hematopoietic cell infiltrates in skeletal muscle (*tibialis anterior*) of LX2931-treated (LX) and control (CTR) mdx pups. Anti-SPL antibody marks SPL in red. CD45 marks hematopoietic cells in green. DAPI labels nuclei in blue. Muscles from two LX2931-treated (LX) and two control (CTR) mice are shown for comparison. (**B**) Quantitation in percentage of CD45+ fluorescent area/low power field of images in A. (**C**) Neutrophil infiltration in skeletal muscle of LX2931-treated (LX) and control (CTR) mdx pups. Neutrophil elastase (NE) marks neutrophils in green. DAPI labels nuclei in blue. (**D**) Quantitation of neutrophil elastase of images in C. Magnification = 400×.

**Figure 5 ijms-23-07579-f005:**
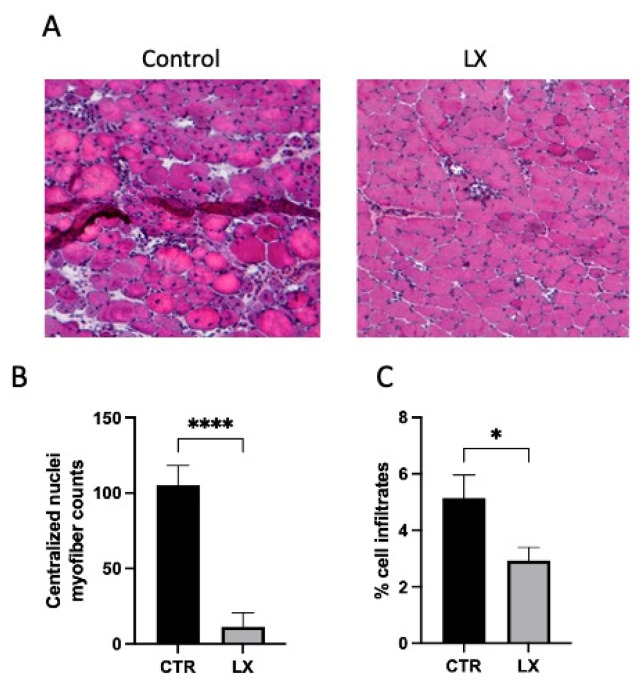
Perinatal SPL inhibition reduces pathological signs of muscle injury. (**A**) Representative hematoxylin and eosin stained cryosections of skeletal muscles (*tibialis anterior*) isolated from LX2931-treated (LX) and control (CTR) mdx pups showing muscle damage is reduced in treated pups. (**B**) Quantification of myofibers containing centralized nuclei. (**C**) Quantification as percentage of cell infiltrates. Shown are mean ± SEM; * *p* < 0.05, **** *p* < 0.001; *n* = 8–11/group.

**Figure 6 ijms-23-07579-f006:**
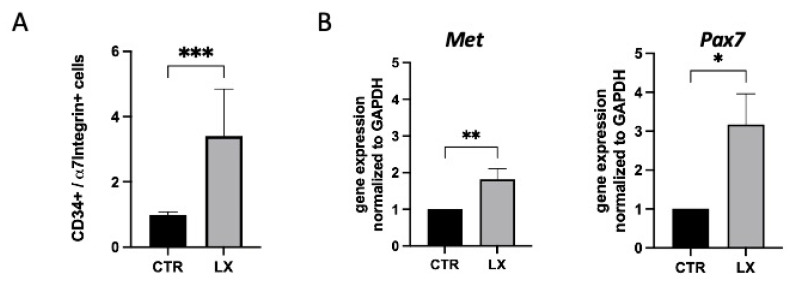
Perinatal SPL inhibition expands the endogenous satellite cell (SC) population. (**A**) Flow cytometry quantification of SCs isolated from disaggregated muscles (*gastrocnemius*) of control (CTR, *n* = 8) and LX2931-treated (LX, *n* = 4) mdx pups. Cells were stained using five selected antibodies. Results were analyzed using FlowJo software. Cells were considered to represent SCs if they were positive for CD34 and a7 integrin and negative for CD31, CD45 and Sca-1. Mean ± SD; *** *p* < 0.005. (**B**) Gene expression of SC markers *Pax7* and *Met* were detected by qRT-PCR and normalized to *Gapdh*. *n* = 4–5/group, and each sample was run in triplicate. Mean ± SD; * *p* < 0.05, ** *p*<0.01.

**Figure 7 ijms-23-07579-f007:**
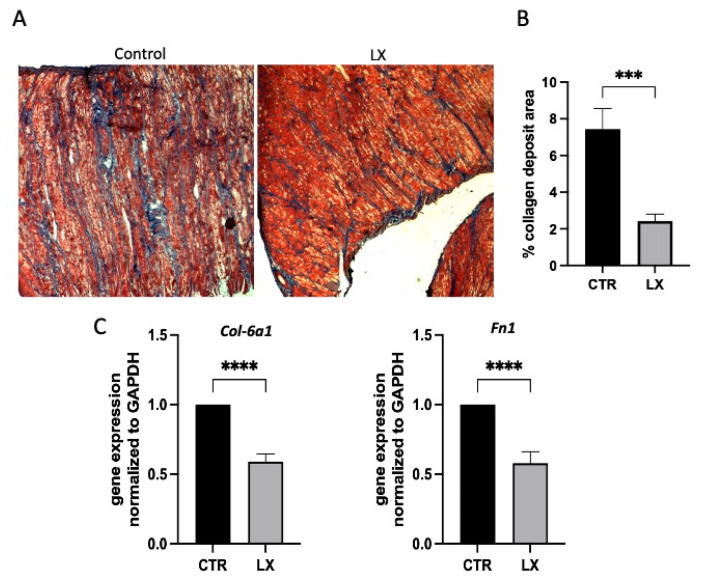
Perinatal SPL inhibition reduces collagen deposition in skeletal muscles of mdx mice. (**A**) Representative light microscopic images of collagen deposition as shown by Masson’s trichrome staining in diaphragms of control and LX2931-treated pups (LX). *n* = 4–5/group. (**B**) Quantitation in percentage of trichrome-stained areas shown in (**A**). Mean ± SD; *** *p* < 0.005. (**C**) Expression of collagen-related genes by qRT-PCR and normalized to *Gapdh*. *n* = 4–5/group, and each sample was run in triplicate. Mean ± SD; **** *p* < 0.001.

**Figure 8 ijms-23-07579-f008:**
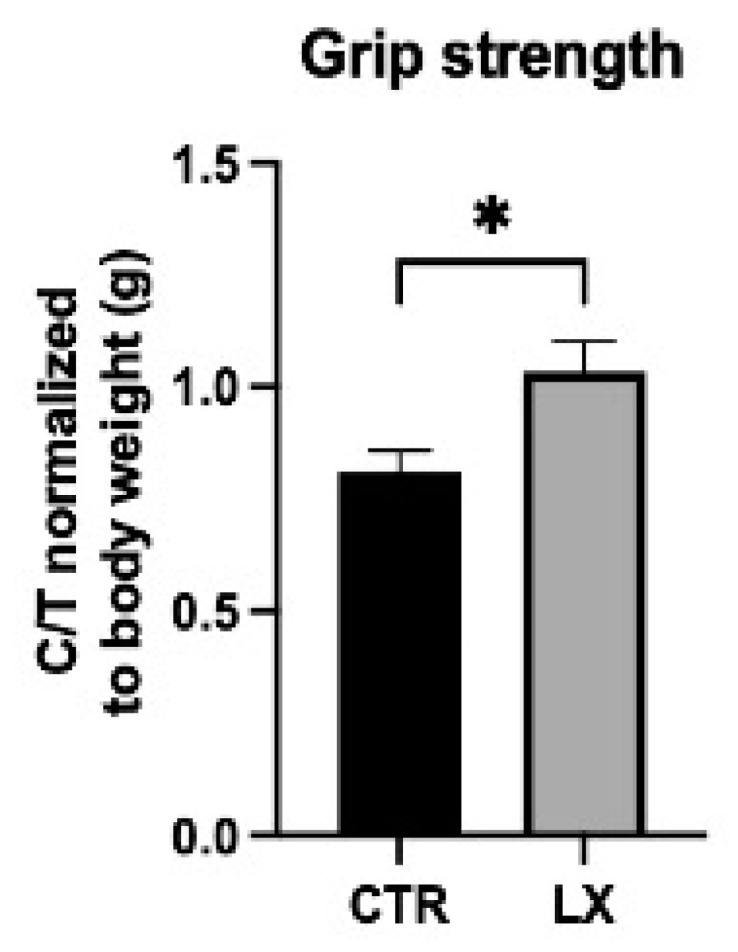
SPL Inhibition improves skeletal muscle function. Grip strength of vehicle-treated (control, CTR) and LX2931-treated (LX) mdx pups at 6 weeks of age measured using a grip strength meter. C/T = compression/tension. Values are normalized to body weight in grams (g). Mean ± SD; * *p* < 0.05; *n* = 8–10/group.

**Figure 9 ijms-23-07579-f009:**
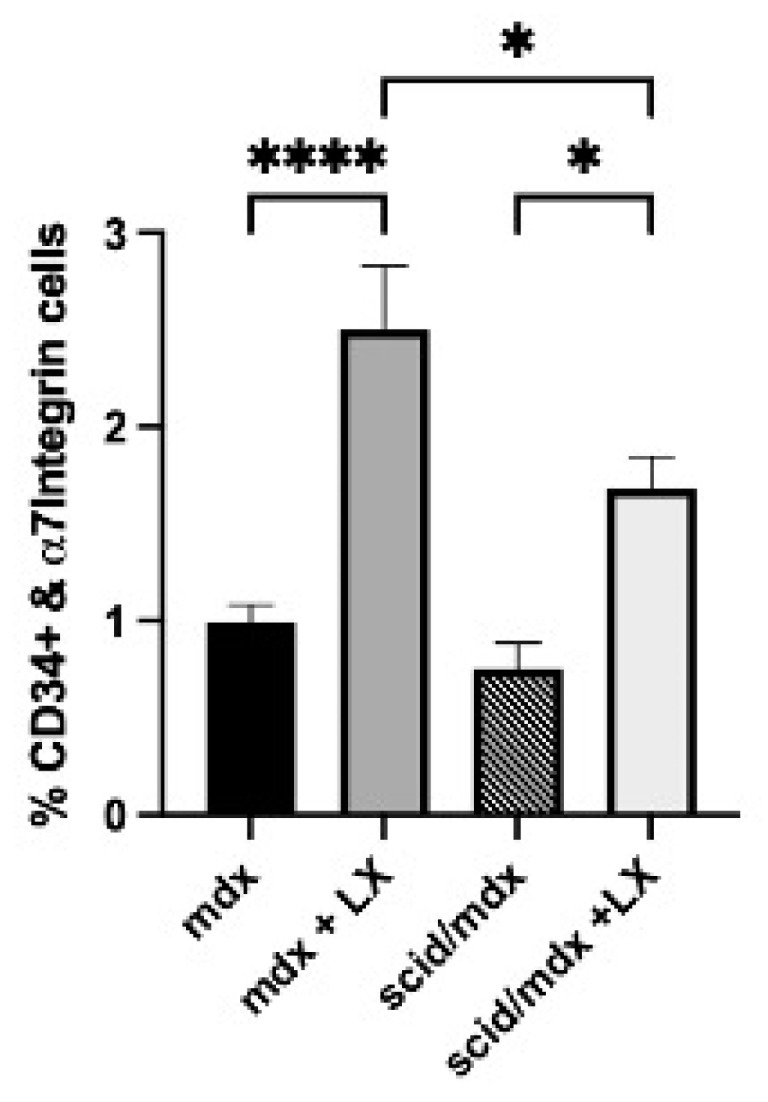
Perinatal LX2931 treatment results in satellite cell expansion in SCID/mdx mice. Female SCID/mdx and mdx mice were treated with LX2931 using the perinatal treatment scheme. Pups were euthanized at three weeks of age, and SCs were quantified in *gastrocnemius* skeletal muscle by flow cytometry as in Figure 6A. Although the absolute number of SCs was significantly lower in LX2931-treated SCID/mdx compared to LX2931-treated mdx mice, the fold elevation was similar in both strains (2.5-fold in mdx and 2.2-fold in SCID/mdx). * *p* < 0.05, **** *p* < 0.001, *n* = 4–5/group.

**Figure 10 ijms-23-07579-f010:**
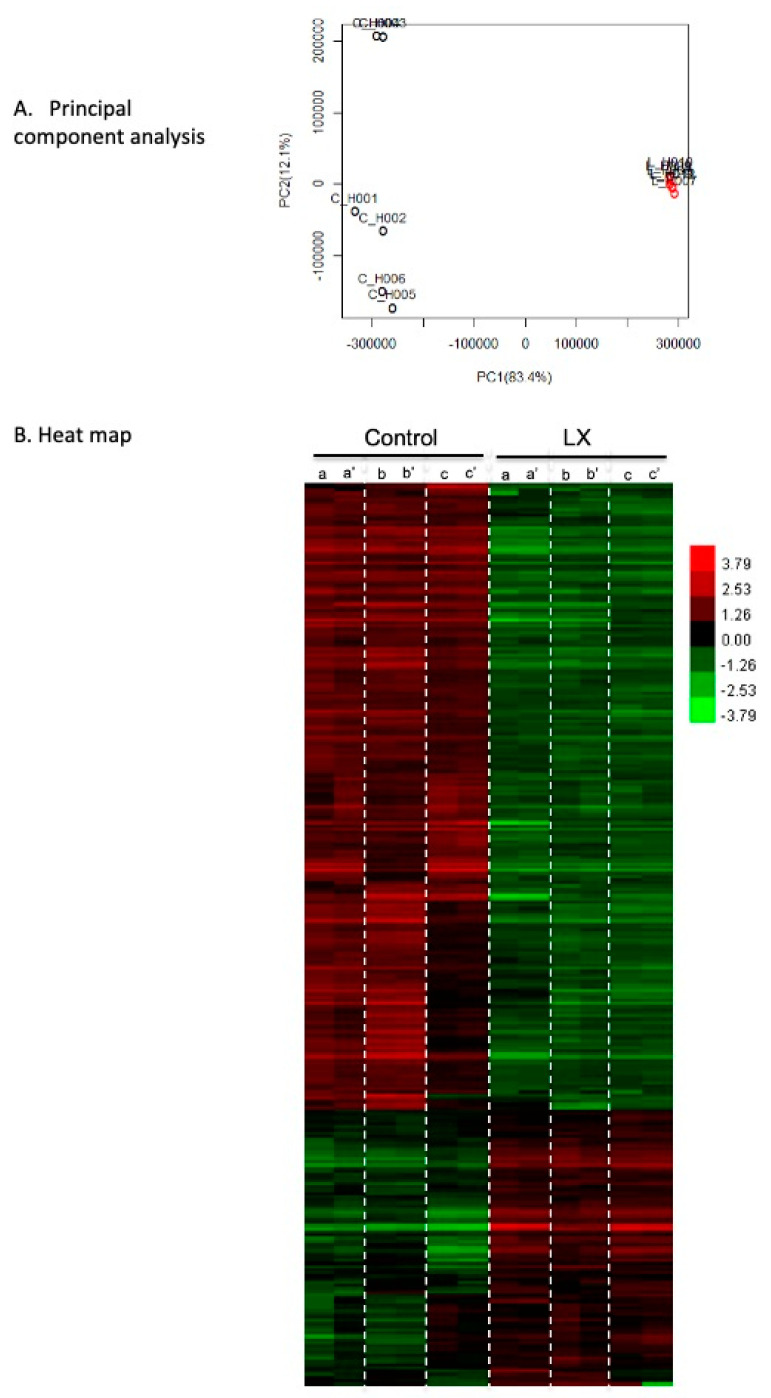
Dystrophic skeletal muscle gene expression profile is significantly altered by LX2931 treatment. (**A**) Principal component analysis of microarray comparison of gene expression in *tibialis anterior* muscles of three control and three LX2931-treated mdx mice run in duplicates. The variable of the first three principal components (PC1, PC2 and PC3) were 83.4, 12.1 and 2.2%, respectively. Note the uniformity of the LX2931-treated expression pattern. (**B**) Heat map of microarray results from 285 genes identified as differentially expressed between control and LX2931-treated (LX) mdx mice, with high expression shown in red and low expression shown in green. The majority of differentially expressed genes in the treatment group were downregulated compared to the control group.

**Table 1 ijms-23-07579-t001:** Top 10 Enriched pathways.

Gene Set Name	# Genes in Overlap (k)	*p* Value
Leukocyte trans-endothelial migration	25	5.1040 × 10^−7^
ECM-receptor interaction	19	4.7618 × 10^−6^
Chemokine signaling pathway	29	1.6142 × 10^−5^
Fc gamma R-mediated phagocytosis	19	5.2623 × 10^−5^
Hematopoietic cell lineage	17	8.7064 × 10^−5^
Cell adhesion molecules (CAMs)	23	0.0004
B cell receptor signaling pathway	15	0.0006
Natural killer cell mediated cytotoxicity	19	0.0009
Focal adhesion	26	0.0011
Axon guidance	19	0.0021

**Table 2 ijms-23-07579-t002:** Top ten molecular functions from Gene Ontology (GO) analysis.

Gene Set Name	# Genes in Overlap (k)	*p* Value
Calcium ion binding	83	7.1745 × 10^−7^
Carbohydrate binding	40	3.9976 × 10^−6^
GTPase regulator activity	41	3.6579 × 10^−5^
Nucleoside-triphosphatase regulator activity	41	5.3041 × 10^−5^
Enzyme activator activity	31	7.4973 × 10^−5^
Guanyl-nucleotide exchange factor activity	22	8.1547 × 10^−5^
Polysaccharide binding	20	0.0001
Pattern binding	20	0.0001
Identical protein binding	33	0.0002
Protein complex binding	14	0.0004

**Table 3 ijms-23-07579-t003:** Top ten biological processes from Gene Ontology (GO) analysis.

Gene Set Name	# Genes in Overlap (k)	*p* Value
Immune response	76	3.0191 × 10^−16^
Cell activation	50	8.3240 × 10^−15^
Cell adhesion	78	3.7950 × 10^−13^
Biological adhesion	78	4.1752 × 10^−13^
Immune effector process	32	2.4120 × 10^−12^
Leukocyte activation	43	2.6950 × 10^−12^
Leucocyte mediated immunity	25	9.2975 × 10^−11^
Inflammatory response	41	1.0399 × 10^−10^
Response to wounding	53	1.2520 × 10^−10^
Vasculature development	43	2.2401 × 10^−10^

**Table 4 ijms-23-07579-t004:** Top ten cellular components from Gene Ontology (GO) analysis.

Gene Set Name	# Genes in Overlap (k)	*p* Value
Cell surface	52	3.2687 × 10^−11^
Proteinaceous extracellular matrix	50	1.3109 × 10^−10^
Extracellular region	165	1.5664 × 10^−10^
Extracellular matrix	51	1.7262 × 10^−10^
External side of plasma membrane	37	8.2316 × 10^−9^
Extracellular region part	85	1.1612 × 10^−7^
Extracellular matrix part	19	9.0239 × 10^−6^
Plasma membrane	228	9.9907 × 10^−6^
Membrane raft	15	0.0006
Collagen	7	0.0007

**Table 5 ijms-23-07579-t005:** Primer sequences used for qRT-PCR.

Gene	Forward Primer (5′ to 3′)	Reverse Primer (5′ to 3′)	bp
*hSGPL1*	CGTGGTCAAGTTGGAGGTCT	ATATAAGAGGGTACTGCCAGCG	128
*mSgpl1*	GGGAAAGTGTGAGATAGCAAG	CTGAGGGAACACGGTACATAAC	94
*Met*	GGCCCAGCTGTTTCAGTGA	CAGCATCGCTCAAATTCAGAGA	61
*Pax7*	CCGAGTGCTCAGAATCAA	ATGCTGTGTTTGGCTTTC	93
*Myf5*	CCAGCCCCACCTCCAACT	CTTTTATCTGCAGCACATGCATTT	127
*MyoD*	AAATCGCATTGGGGTTTGAG	GAGCGCATCTCCACAGACAG	178
*Col-1a1*	TCCGGCTCCTGCTCCTCTTA	GTATGCAGCTGACTTCAGGGATGT	78
*Col-3a1*	GCCCACAGCCTTCTACAC	CCAGGGTCACCATTTCTC	109
*Col-6a1*	GATGAGGGTGAAGTGGGAG	CACTCACAGCAGGAGCACAT	185
*Fn1*	TGCCTCGGGAATGGAAAG	ATGGTAGGTCTTCCCATCGTCATA	78
*hGAPDH*	AATCCCATCACCATCTTCCAG	AAATGAGCCCCAGCCTTC	122
*mGapdh*	ACCTGCCAAGTATGATGA	GGAGTTGCTGTTGAAGTC	118

bp, base pairs; hSGPL1, human SGPL1; mSgpl1, mouse Sgpl1; hGAPDH, human GAPDH; mGapdh, mouse Gapdh.

## Data Availability

The complete transcriptional profiling data set and GO analysis is publicly available at the Mendeley Data website using the link http://dx.doi.org/10.17632/mw4syvnmgw.1. All other data generated in this study are presented in the main text and Appendix A of the manuscript.

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
