# Peer review of "Sphingosine Phosphate Lyase Is Upregulated in Duchenne Muscular Dystrophy, and Its Inhibition Early in Life Attenuates Inflammation and Dystrophy in Mdx Mice"

_ijms, 2022, doi:10.3390/ijms23147579_

Round 1
Reviewer 1 Report
The manuscript is well-writen. Some minor improvemnets required:
1. Please expand the Discussion section, be adding more informantion for possible theurapeutic implications of the finding.
2. Describe statistical analysis method in more detail.
3. Minor editing for typos
Author Response
We thank Reviewer 1 for his/her helpful comments and have revised our manuscript accordingly. Specifically, we have expanded the Introduction section and included a new paragraph at the end of the Discussion section expounding on the relevance of our findings to the potential treatment of DMD. We have also incorporated many additional references in these sections. We have provided more details in the Method section, specifically in relation to muscle types used, the methods we employed for establishing significance, and other important information. We have reviewed the manuscript carefully for typographical errors and tried to be more uniform in the use of hyphenated terms. We again thank our reviewer for taking the time to critique our manuscript and hope they will find this version acceptable for publication.
Reviewer 2 Report
I have significant concerns. The paper uses the muscle of mdx mice for all of its studies. In the methods section, it mentions that three types of muscle are used in the study--gastrocnemius, tibialis and diaphragm. In several places in the paper the muscle used for the analysis was mentioned, but sometimes the muscle is simply referred to as "skeletal muscle". This is not appropriate, because these three skeletal muscles are as different as night and day in this animal disease model and should not be lumped together. In a paper (https://www.ncbi.nlm.nih.gov/pmc/articles/PMC4301874/) that produced detailed quantification of these muscles in mdx mice at different ages, the results were wildly different from the quantifications cited here. For instance, in that paper, they looked at 8 week-old gastrocnemius mdx muscle saying the following: "Overall, quantification of collagen content indicates that fibrosis is absent in wild-type and mdx GM muscle" (Figure 3I). In fact, they were virtually identical levels of fibrosis at about 3%. In 8 week-old diaphragm, they did see a trend towards more fibrosis in the mdx muscle, 11% vs. 7%, but not statistically significant, just a trend. In this paper we are reviewing however, in Figure 7, they show the mdx untreated mice at 7.5% vs. 2% in the treated mice. The muscle type is not mentioned. The mice ages are the same. So the results just are incongruous. It cannot be gastrocnemius, because otherwise the collagen should be more like 3% instead of 7.5%. Diaphragm is possible (7.5% is not too far from 11%) but the drop to 2% which is less than 1/3 of the collagen level of that PLOS One paper's control diaphragm just seems unlikely. Then instead of doing Western blots to quantify it, RT-PCR was performed, which is not as convincing. Last but not least, in none of the experiments were real non-dystrophic controls included. This makes most of the interpretation more difficult to make.
Author Response
We thank Reviewer 2 for taking the time to critique our manuscript and for the helpful feedback they have provided. We have taken their concerns to heart and revised our manuscript accordingly. Specifically, we have:
1) Substantially increased the number of references in the Introduction, Discussion and Methods sections of the manuscript, adding more clarification in the Methods section
2) Rewritten certain sections to make the intent and specificity of our methods more clear to the reader
3) Specified which muscle group was used in every experiment. We wish to point out that unlike in adult mice, muscle tissue is limiting in studies of juvenile mice. Our analysis of S1P in muscles of diaphragm and TA groups showed similar trends. Therefore, we tried to use the tissues to gain the most information from each animal subject, always comparing the same muscle group between treated and untreated mdx, while recognizing that muscle groups can be very different in some respects. Diaphragm was used for all fibrosis studies due to the greater sensitivity to fibrosis as indicated by the reviewer and the referenced study. TA was used for all microarray analyses. This information is now provided in the text, figure legends, methods sections and is described as a limitation in the discussion section of the revised manuscript.
4) We have provided a more nuanced interpretation of the fibrosis aspect of the study in the Discussion section of our revised manuscript, citing the article mentioned by the reviewer and the caveats that have been raised. Despite these caveats, we feel that the cumulative observations we have made support our study's main conclusions.
5) Non-dystrophic mice were included in the initial experiments of our study in which we characterized sphingolipid metabolic changes over time in young mice (mdx and non-dystrophic controls). However, thereafter we did not include non-dystrophic controls. This decision was made because we were most interested in exploring the impact of the perinatal delivery of LX2931 and raising of muscle S1P levels on the well-characterized features of mdx mice. This comparison is similar to how any randomized clinical trial would be conducted, i.e., with a placebo and an intervention. These are also now listed as limitations of our study in the discussion section.
6) We chose to perform qRT-PCR rather than western blot to evaluate expression of key targets because of the limiting amount of muscle tissue in order to gain as much information as possible. In the future, it will be important to confirm findings obtained through qRT-PCR by western blotting or other methods.
7) Lastly, we expanded our discussion of the transcriptome analysis to explain why our focus was on comparing drug treated vs. untreated mdx groups, namely that the primary impact on the transcriptome of the mdx genotype is immunological, and thus we hypothesized that the immunomodulatory regulator LX2931 would reverse this phenotype.
In summary, as guided by our reviewer, we have made significant changes to improve the manuscript, including a comparison of our findings with those of published reports, providing some more nuanced language in the interpretation of fibrosis models, and an explanation of all muscle types used in every experiment. We appreciate the time taken to review this study and hope that it is now acceptable for publication.